# Stockholm Paradigm in the Study of Influenza H1N1 Viruses: A New Approach to the Study of Zoonotic Risk Coupling Multiple Correspondence Analysis and Multi-Locus Phylogenies

**DOI:** 10.3390/v17101350

**Published:** 2025-10-08

**Authors:** Sofia Galvão Feronato, Rafael Antunes Baggio, Hellen Geremias Gatica Santos, Guilherme Ferreira Silveira

**Affiliations:** 1Grupo de Imunologia Molecular, Celular e Inteligência Artificial, Instituto Carlos Chagas, Fiocruz/PR, Curitiba 81350-010, Brazil; sofia.feronato@aluno.fiocruz.br; 2CTPeixes, Instituto de Ciências Biológicas, Universidade Federal de Minas Gerais, Belo Horizonte 31270-901, Brazil; rbaggioufpr@gmail.com; 3Instituto Carlos Chagas, Fundação Oswaldo Cruz, Curitiba 81350-010, Brazil; hellengeremias@gmail.com

**Keywords:** emerging infectious diseases, Influenza A H1N1, Stockholm Paradigm, machine learning, zoonosis

## Abstract

The Stockholm Paradigm, a multilevel framework for studying coevolutionary interactions, it is a promising method for obtaining a globally relevant understanding of the emergence of present and past host–parasite and insect–plant interactions. This research aimed to expand the application of the Paradigm to virus–host interactions, considering that viruses are being subjected to the same evolutionary forces as any other living organism. By applying different data science techniques, we described and discussed capacity and opportunity traits for Influenza A H1N1 strains, and how they might influence the pathogen’s host repertoire evolution, and thus ranked different strains according to their emergence risk in the human population. We hope to contribute to the application of different methods for understanding disease emergence, and consequently to the development of new public health strategies for preventing (re)emerging diseases.

## 1. Introduction

The Stockholm Paradigm aims to recover Darwinian ideas in order to discuss evolution, claiming that the historical construction of Neodarwinism lost essential information on the study of diversification of life [1,2,3]. The authors who proposed this Paradigm aimed to discuss the coevolution and emergence of diseases as processes that imply the synergy of capacity and opportunity in determining biological interactions.

Capacity is defined as the whole ability of an organism to utilize resources, either as a free-living agent or as a symbiont, given by the Information Space inherited by their history, including actual genetic, epigenetic and developmental information, as well as information acquired by other means, such as education. On the other hand, opportunity is defined as the coincidence of resource availability (either environmental or host-wise) in time and space; thus, the realization of capacity is conditioned by the opportunity [4].

The authors extensively discuss the framework underlying the Stockholm Paradigm, which builds upon earlier Darwinian proposals and challenges the general Neodarwinian view of the one host–one pathogen relationship. It relies on three general pillars: (1) ecological fitting [5,6,7] (2) oscillation hypothesis [8], and (3) taxon pulse [9,10]. Ecological fitting approaches the initial stages of an interaction, describing the interaction in terms of realization of capacity, limited opportunity, selection of phenotypic variations, and establishment of an interaction.

Oscillation hypothesis and taxon pulse, on the other hand, address the phylogenetic and phylogeographic aspects of evolution. The former explores the transitional aspect of specialism and generalism, while the latter describes the spatiotemporal reality of evolution. In particular, it highlights the exploration of hosts by parasites, for example, by cycles of stability and perturbation, or better yet, through the oscillation of opportunity. The Stockholm Paradigm aims, therefore, to present evolution from populational to the phylogeographic perspectives [1,9,10].

The Stockholm Paradigm had not been applied to host–virus coevolutionary scenarios prior to the emergence COVID-19 [11] However, it has been extensively analyzed in the study of other infectious diseases. Therefore, this research aimed to expand the study of viral coevolution with its hosts under the scope of the Stockholm Paradigm by applying a new methodology and evaluating whether it can help understand and draw expectations on host-switching events in virus–host interactions.

In order to achieve this, one needs to choose a viral lineage that can be deeply analyzed according to capacity and opportunity, and to gather sufficient data on both parameters. Also, the viral lineage needs to be one that has been sampled, studied and monitored in depth and over a long period of time.

With that in mind, we opted to use Influenza A H1N1 virus lineage as a study model, since it has a long-lasting relationship with the human population (Figure 1), and has been under the scope of the global public health since the beginning of the 20th century, marked by the 1918 Spanish flu [12], followed by several epidemic and pandemic events, with a vast host oscillation among avian and mammalian taxa [13].

For these reasons, Influenza H1N1 carries a great amount of public data that can fit into the concepts of viral capacity and transmission opportunity, and better discuss the evolutionary patterns and expectations the Paradigm presents, enabling a better understanding of host–parasite coevolution, herein designated as host–Influenza coevolution, in the field of disease emergence risk assessment, as has been previously discussed for other diseases [14,15,16].

It is important to note, however, that H1N1 was chosen as a study model and not as the sole model for this application. Our main purpose here is to develop a method for zoonotic prediction of Influenza viruses in a broader sense.

This work relies heavily on the assumptions of ecological fitting, where capacity and opportunity conditions precede and enable the emergence of new symbiont interactions. The idea here is to describe the capacity Influenza A H1N1 strains have to establish themselves in human populations by analyzing their epitope and substitution profile, coupled with the current ecological circumstances (available surveillance data) that allow this. By analyzing both capacity and opportunity, we expect to be able to visualize, and hierarchize, the sampled strains according to their emergence risk in humans, contributing to emerging infectious diseases preparedness.

## 2. Materials and Methods

In order to fulfill what is described by the Stockholm Paradigm—a synergistic view of all of the possible capacity information available for each strain as well as the opportunity scenario involved in an interaction—we needed to create a method which would allow us to analyze all these parameters at the same time, with no predetermined weight, direction or hierarchy to the information.

To obtain this global understanding of each strain, we applied the unsupervised learning method named Multiple Correspondence Analysis. Multiple Correspondence Analysis (MCA) is a dimensionality reduction method [17] which allows us to obtain just what the Paradigm requires: an unbiased understanding of the different capacities of each strain. It also allows us to confront that information with the ecological context of the strain. By understanding the capacity variation in a group of strains, we expect to rank them according to their compatibility with the human host (using the Hierarchical clustering analysis—HCA), and filter, among those with capacity to use the human host, the strains with the spatial and temporal opportunities to do so, which would allow them to actually emerge in our species.

We will first describe the capacity and opportunity data we were able to collect from the Influenza Research Database (IRD) deposited in the BACTERIAL AND VIRAL BIOINFORMATICS RESOURCE CENTER v.3.30.19 (BV-BRC), and we will then discuss the data preprocessing and processing methods, followed by the statistical application of the aforementioned MCA.

### 2.1. Raw Data Collection

All the data used in this work was collected from the Influenza Research Database (IRD) [18] which is currently part of the BACTERIAL AND VIRAL BIOINFORMATICS RESOURCE CENTER v.3.30.19 (BV-BRC), a larger database that includes all bacterial and viral research databases, as well as archeas and eukaryotic hosts. The current database version can be accessed at BV-BRC [19].

The data that describe viruses’ capacity to utilize their host will be herein described as (1) strain information; (2) epitope information and (3) substitutions information—or known genomic substitutions that are associated with capacity alterations, such as transmission capacity, replication change, infectivity in a host, and antiviral resistance, among other specifications. Opportunity will be described by (4) surveillance information, which includes variables that describe the ecological context of different H1N1 strains.

The whole preprocessing, processing and Multiple Correspondence Analysis (MCA were performed using the Jupyterlab platform, Python version 3.9, while Hierarchical clustering analysis (HCA) was applied thought Google Colaboratory in order to provide access to some specific dependencies.

### 2.2. Data Importation

All the raw data collected from IRD was manually imported by filtering the database information for Influenza A/H1N1/avian, human, or mammalian hosts. Meanwhile, the genomic information utilized in the Epitope Occurrence section was collected by remotely accessing Genbank with the *Entrez* and *SegIO* Python libraries, which locate and collect the genomic information through the *Genbank ID*, delivering the nucleotide and aminoacidic sequences of all segments analyzed in the **string** format. See the Entrez documentation, as part of the Biopython package (Bio.Entrez).

The dataset used to characterize viral capacity include (1) strain information—with strain name and GenBank ID for the 8 genome segments; (2) epitope information—with information on all linear peptides registered in the BV-BRC; and (3) substitution information—corresponding to all human, avian and mammalian genomic substitutions that have been identified by published works and are described as substitutions that alter viruses’ capacity to interact with their host (published in PubMed). Capacity information was processed and evaluated using MCA and HCA.

In order to describe the opportunity for H1N1 strains to emerge in our species, we collected surveillance information from the original BV-BRC dataset. This dataset includes all information on data collection (date and location in terms of coordinates, city, state/province, and country), and host identification and condition (in terms of natural state, capture mode, and health at the time of sampling).

The raw version of this data was obtained in January 2023 through the following link (Raw Strain Data IRD, [19]).

### 2.3. Data Processing Protocol

We imported each preprocessed dataset as will be described—and quantified—below (the method workflow is described in Figure 2).

For the strain information, by the end of the data processing pipeline, we obtained 18,045 H1N1 strains, with the GenBank ID for all 8 segments of the strain. For the epitope information, we obtained the genomic sequence of each of the 8 segments of the 18,045 strains and translated it into the aminoacidic sequence for detection of the 8595 epitopes given by the BV_BRC database. We then filtered the detected epitopes with a 5% occurrence threshold in order to avoid low-frequency epitopes that could bias our analysis. By the end of this process, we had imported 6488 epitopes into our MCA.

In order to process the substitution information, we imported each substitution subgroup (avian, human and mammalian) and merged them separately because we did not require that the *Strain* present in the avian substitution table, for example, be present in the human or mammalian tables. It is important to note that when we merged both epitope and substitution datasets, we identified the following Influenza strain host groups: of the 18,045 strains, 3625 of them were non-human mammalian strains, alongside 13,335 human and 608 avian strains, totaling 17,568 strains. The remaining strains (of the 18,045 originally imported in the strain information) were not present in the substitution table and were excluded from the analysis.

The final dataset—herein referred to as the Capacity Table—included 17,568 strains and 6521 columns which correspond to the cumulative capacity data and the strains that are present on both datasets (Genbank ID, epitope occurrence and substitution data). This information was the input used in our subsequent analyses.

### 2.4. Statistical Analysis

#### 2.4.1. Multiple Correspondence Analysis (MCA) Method

We performed MCA on the Capacity Table in order to derive a set of continuous and uncorrelated Principal Components (PCs), which summarizes the information of all parameters at once, in the least number of components as possible [17].

MCA is an unsupervised learning method suitable for reducing the dimension of data tables represented by individuals and their answers to a set of categorical variables. It can be applied to study individuals, variables, or categories. We applied MCA to the study of parameter categories to be able to represent Influenza strains according to the commonness of their parameter states (or commonness of capacity information), which was represented by a cloud of category points in a Euclidean space [17].

MCA interpretation is based on the estimated *eigenvalues* and *eigenvectors*. The former are positive quantities related to inertias, representing the total amount of explained variability (percentage of inertia) by a given PC—the information we utilize in order to consider how well a specific PC explained the data points’ variability in the Euclidean space. The *eigenvalues* order the PCs from the one that most explains the variability of the data to the one that explains the least [20]. So, we retained for HCA those PCs that accumulated approximately 65% of explained variability in order to adequately identify subgroups among the Influenza strains that might have biological significance in the identification of potential zoonotic H1N1. *Eigenvectors* represent the orientation of the PCs.

#### 2.4.2. Hierarchical Clustering Analysis (HCA)

For HCA, we used the Agglomerative Hierarchical Clustering (AHC) method to calculate individual similarity using Ward’s method and Euclidean distance, resulting in a hierarchical tree, which segregates the Influenza strains from most similar to most dissimilar strains distributed in the Euclidean space (also known as the *bottom-up* dendrogram construction method described by [21].

The number of clusters was initially determined by visual inspection of the hierarchical tree. Additionally, we considered the increase in within-cluster homogeneity, represented by an inertia plot, and cluster interpretability, according to Influenza strain information, that might contribute to cluster identification and standardization.

#### 2.4.3. Epitope Profile Characterization

Following cluster identification, we analyzed the distribution of epitope occurrences to identify key differences among the groupings formed by HCA. For this purpose, we calculated the relative risk (RR), which simply constituted the relative frequency of the epitope in a cluster in relation to its relative frequency in the cluster of reference, for example, cluster 1:$$RR\;35\;x\;36\;=\;\frac{Relative\;frequency\;of\;data\;in\;cluster\;2}{Relative\;frequency\;of\;data\;in\;cluster\;1\;}\;\mathrm{and}\;RR\;37\;x\;36\;=\;\frac{Relative\;frequency\;of\;data\;in\;cluster\;3}{Relative\;frequency\;of\;data\;in\;cluster\;1}$$

#### 2.4.4. Surveillance Data Processing

The surveillance data was analyzed only after the capacity clusters were explored, and was not used as input for the MCA and HCA. Of the 115,576 surveillance events gathered from the BV-BRC database, 48,159 involved a strain name present in the capacity data and were therefore considered for subsequent analyses.

We were interested in information that depicted any aspect of host interaction *opportunity*, or the ecological setting of the different Influenza strains. Thus, we processed and organized the surveillance data according to the identified strain and retained the following information of potential interest: for steps i and j of the workflow, we labeled and grouped the data as (i) *Dados_host_location,* which included host species, host common name, host group, and host natural state; and (j) *Dados_host_location*, which included collection year, collection country, collection state, and collection city. The final surveillance information was separated into two datasets: (1) host nature data—describing the host species and group, as well as nature (domestic or wild). Unfortunately, only the host group contained a sufficient amount of data. Of the 17,568 strains included in the clustering analysis, only 1218 strains had host nature information (this is step ***i*** of the method workflow described in Figure 1 and Figure 2 host location data—including city, state and country information. We manually searched for all the location specifications, so as to enable us to subsequently relate locations based on regional congruence (this is step ***j*** of the method workflow described in Figure 1). Both datasets were then filed to be used as the *opportunity* information of the selected strains.

In order to try and increase the number of strains analyzed in terms of host use, we applied a second method to observe the host group distribution according to the capacity clusters, which we referred to as the Strain Name Segregation Method (Appendix A).

This additional method is applied using the crosstab files, which are the outputs of the spreading technique described in the Appendix A. This application allowed us to observe the strain distribution across host groups for 4210 strains, but did not enable a wider discussion about the other surveillance information, unlike the main result presented here.

### 2.5. Phylogenetic Reconstruction

#### 2.5.1. Sampling Representative Strains

The complete set of strains that were included in the MCA and HCA rounded up to 17,521 strains. Ideally, we would construct our phylogenies using all sequences and strains; however, for this number of strains, such an approach was time-consuming and computationally challenging. Thus, we selected the 30 most representative strains of each cluster using the following criteria: first, we separated, for each of the three clusters, all epitopes which occurred at a frequency above 50%, among the epitope possibilities.

After that, we arbitrarily pulled the top 30 strains with the highest frequency of these epitopes to represent the cluster epitope profile. Not all selected strains carry all the high-frequency epitopes of the cluster, but they are the ones with the highest number of them. We then used the Strain Table in order to pull all amino acid sequences from the sampled strains and construct the trees, beginning with the hemagglutinin segment construction.

#### 2.5.2. Why So Many Trees?

When analyzing the evolution of an Influenza group, one must consider the different ways we can draw the phylogenetic trees. A group is traditionally understood according to one gene, which is considered the most representative of the taxon, since it requires less information in order to be drawn and the method is less time-consuming. However, one of our goals here was to test the different ways we can draw the tree using one or more genes.

After applying MCA and HCA we decided to construct two trees with the highest cluster differentiation power—the hemagglutinin (HA) and the Nucleoprotein (NP) segments—and see how they resolved the phylogenetic relationships among the strains in comparison to the capacity clusters. After that, we drew a two-gene tree containing the two segments and compared it to a full genome tree (with the eight segments of each strain). This comparison was important for helping us understand the distinguishing power of the two most significant segments (according to our method) compared to the full genome.

#### 2.5.3. Phylogenetic Procedure

In order to explore the phylogenetic relationship of the selected Influenza A H1N1 strains, we followed the traditional pathway for Bayesian Phylogenetic Application using the tool StarBeast package in the BEAST2 (v.2.7.6).

We first collected the *fasta* sequences using the Python library Entrez for all genome segments of interest. In order to generate a reliable phylogeny, we then (1) aligned the sequenced with *Muscle* and trimmed all the amino acid sequences using Geneious (v. 7.1.3) [22]; (2) defined the best-fit substitution model for all sets of sequences (single-gene, two-segment, and later, eight-segment reconstructions) using MEGA X [23]; (3) created the BEAST2 [24] input in the BEAUTi (v.2.7.6); and (4) ran a Bayesian analysis using BEAST with a run with 50 million MCMC, sampled every 1000 generations, using the JTT+G substitution model.

The phylogenetic analysis was assessed in Tracer (v.1.7.2) [24], which allowed us to determine the Estimated Sample Size (ESS) of each tree metric, and thus comprehend the general quality of the parameter landscape estimations. The trees were run in TreeAnnotator (v.1.10.4) [24] to remove the burn-in (10% of the sampled MCMC) and create a consensus regarding the tree topology. The final tree was drawn using FigTree (v.1.4.4). Further editing of the tree was manually performed using Canva software v. of 24 April 2024.

This multi-gene construction is only possible using the newest versions of BEAST2 [24], because it contains the *Beast template that allows us to add to the workspace-independent gene evolutionary scenarios, drawing a consensus tree from them. This variation was applied both to the HA+NP phylogeny and the 8-segment reconstruction, where we tested all possible MCMC models available in the software (Yule, calibrated Yule, FBD, and birth–death) for the first multi-locus reconstruction and analyzed their likelihood using Tracer. Since the most adequate model was the FBD for the two-segment tree, we kept the same model in order to analyze the 8-segment construction.

#### 2.5.4. Phylogenetic Signal Analysis

This application was later utilized to analyze two characters in the resulting trees: (1) HCA cluster phylogenetic signal and (2) host use phylogenetic signal, both of which were constructed using the Mesquite software (v. 3.61) [25]. This statistical analysis explains in a more systematic way whether the character-state transformation along the tree is highly associated with the group evolution or whether it is distributed randomly along the lineages. The phylogenetic signal can be seen as a tool for visualizing the association between strain and character evolution. A null model was created by reshuffling the terminal taxa of the tree 1000 times to assess the significance of the phylogenetic signal.

## 3. Results

### 3.1. Multiple Correspondence Analysis (MCA)

From the final capacity dataset, with 17,568 strains and 6521 columns (epitope and substitution processed data) we applied the MCA in order to derive a set of continuous and uncorrelated PCs. The resulting MCA (Figure 3) was constructed using five PCs, which cumulatively explained 60% of the variability of the data (Figure 3 and Table 1).

Following MCA results, we applied an HCA to the retained PCs, using the AHC method in order to identify the Influenza subgroups according to the similarity of general capacity (see the multicolored plot in Figure 3 and Figure 4). Visually, we distinguished three subgroups. Since Influenza strains were not homogeneously distributed in the Euclidean space (Figure 3) nor capacity clusters (Figure 4), they required further characterization.

### 3.2. Cluster Description

The HCA has given us a total of three clusters, where the first cluster contains 2249 strains, the second contains 4979 strains, and the third contains 10,340. In order to begin understanding the general epitope pattern each cluster bears, we identified the general epitope frequency, followed by the relative risk (RR) calculation (Figure 5).

The epitope RR distribution supports the theory that clusters 1 and 3 have extremely distinct epitope profiles compared to the highly diverse cluster 2, which is observed by the homogeneous (and lower) relative risk values in clusters 1 and 3 relative to cluster 2 (Figure 5a,c). In the cluster 3 vs. 1 comparison (Figure 5a), the epitopes present in cluster 3 occur at a very reduced frequency, except for a conglomerate of outliers that are 15 to 20 times more frequent in cluster 3 compared to cluster 1.

Cluster 2 presents a much more diverse epitope relative risk distribution when compared to the other clusters as reference groups Figure 5a,c. This is supported by the frequency distribution in Figure 5a,c, where the quartiles are more amply distributed in the y-axis, indicating a large variation in the epitope frequency when compared to cluster 1 and cluster 3, respectively. Cluster 3 and cluster 1 are less diverse in terms of epitope frequency variation (Figure 5a,c). Finally, to draw a general picture of the cluster profile, we will analyze the epitopes or substitutions which contribute most to the cluster characterization. This can be determined by characterizing the main components of MCA, which will allow us to calculate which epitope contributes most to the strains’ distribution in the Euclidean space (Table 1).

With this metric, we realize that the ten most influential capacity parameters are all epitopes, with the relative risk metric varying from 0.73 up to 23. We drew a barplot to identify possible patterns (Figure 6).

In Figure 5, one can notice how, for the ten most influential epitopes of the first two dimensions, they are completely opposite in terms of epitope profile. For the first five epitopes, cluster 2 retains twenty times more epitope occurrences, while cluster 3 presents approximately 30% less epitope occurrences compared to cluster 1 (RR = 0.73), whereas for the last five epitopes, cluster three retains almost twenty times more epitope occurrences, while cluster two retains six times more epitope occurrences compared to cluster 1.

After determining the relation of the 10 most influential epitopes to each cluster, we explored their genetic location and possible importance for the Influenza strains’ host utilization strategies. The 10 most influential epitope sequences are located in two proteins (and genome segments) of the virus (Figure 6), where Epi 1 through 5 belong to the hemagglutinin (HA) segment—an essential surface protein for host cell recognition and invasion—while Epi 6 through 10 belong to the Nucleoprotein (NP) segment, which structures and envelops the viral genome segments and composes the ribonucleoprotein complex (vRNP) [12].

Considering the cohesion of this information with the RR calculations we previously generated (Figure 5 and Figure 6), we constructed different phylogenies as described in the Phylogenetic Reconstruction Section.

Both HA and NP segment trees (Appendix A) disagree with the HCA construction and have low bootstrap values in different nodes of the tree (the NP tree in the group with a lower resolution), and this can be explained by the fact that the HCA tree considers the 6488 epitopes present in all eight genomic segments of Influenza, indicating that the segment evolution does not converge with the whole genome of the group, even though they are considered the two most influential epitopes. Based on this, we constructed a multi-locus analysis for the same segments given by the MCA (HA and NP segments) to see if this tree would resemble the previously constructed clusters, followed by a multi-locus phylogenetic construction using all eight segments.

### 3.3. Multi-Locus Phylogenetic Constructions

The cluster was highlighted in blue when the majority of the strains belonged to cluster 1, in green when the majority belonged to cluster 2, and in red when the majority belonged to cluster 3 (Figure 7 and Figure 8). For the two-segment reconstruction (Figure 7), the clusters identified in the phylogeny were not absolute matches to the HCA-identified clusters, but they did converge considering the general groupings (reflected by the RR calculations).

It is important to note that, although the general branch bootstrap values for both multi-locus trees were higher compared to the single-gene trees (Figure 7 and Figure 8), they did not completely converge with the so-called capacity hierarchization (Figure 3). They did, however, become more similar in terms of cluster identity. The HA+NP construction (Figure 7) presents low bootstrap values for the intermediate nodes, which indicate low intra-cluster differentiation capacity. However, the two nodes that distinguish the three clusters in Figure 6 do present high posterior values, indicating that the three clusters of interest were in fact identified. The full-genome tree, in turn, presented a high confidence level for almost all notches, with a higher differentiation capacity.

For cluster A—with the majority of strains belonging to cluster 1 of the HCA (highlighted in blue in Figure 7 and Figure 8)—there was an RR of 2.99 for a randomly selected strain of that cluster (Figure 8) belonging to cluster 1 of the HCA, and an RR of 1.83 of the same scenario for the HA+NP tree (Figure 7). For cluster B—with the majority of strains belonging to cluster 2 of the HCA—there was an RR of 1.1 for the strain belonging to cluster 2 in the full-genome tree and an RR of 1.37 in the HA+NP tree. Finally, for cluster C—for which majority of strains belonged to cluster 3 of the HCA—there was an RR of 2.22 of the strain belonging to cluster 3 in the full-genome tree and an RR of 2.25 in the HA+NP tree.

We also applied phylogenetic signal analysis in order to add robustness to our understanding of the trait evolution for both cluster identity and host use.

For the eight-segment phylogeny (Figure 9), the original tree presented 39 character transformations, which were also above the *p*-value mark (38 changes). Therefore, the cluster identity transformation was marginally significant, suggesting the presence of a phylogenetic signal. Figure 9c presents the host use trait evolution, where for the eight-segment tree there were 10 character-state transformations, while the analysis shows that 16 transformations is the number of transformations with a *p* = 0.05. Based on this we can conclude that there is a strong phylogenetic signal for host utilization in the sampled group.

### 3.4. Surveillance Data Concatenation

For the Host Nature dataset, we observed that only 1218 strains of the total 17,568 included in the MCA and HCA actually had surveillance nature data. The processed host names (Figure 10a) and Group (Figure 10b) can be observed in the following plots:

In the host nature dataset, the clusters are not uniformly distributed among the host groups, where cluster 1 is the main group in human strains (above the 80% mark) (Figure 10b), while avian hosts are predominantly located in cluster 3 (this is true of more than 90% of the registered avian strains). Swine hosts carry an almost equal number of strains from cluster 1 and 3, while the strains from cluster 2 are homogeneously distributed among the three host types (Figure 10b)—information which coincides with the group’s high epitope frequency variation and diversity compared to the remaining groups.

We can also observe how the specific host distribution (Figure 10a) is not homogenous among clusters nor among specific hosts. Humans, the American Black Duck, the Common Teal, the Greater-White-Fronted Goose, the Red Necked Grebe, swines, and the Wild Boar were the specific hosts with the highest number of samples, followed by the Mallard and the Pig.

From the host location dataset, we can observe the general distribution of the sampled H1N1 that are included in the clustering analysis from a microregional, local, state and national point of view. We also manually added a *specification* column so as to refer to the specific quality of the microregions (Figure 11) identified. This column includes the variables *Urban*, *Forestal*, *Periurban*, *Brackish*, and *Rural*. We understand that environmental traits do not define the environmental selection of the Influenza strain; however, they define the hosts’ (and potential hosts’) spatial distribution.

After analyzing the surveillance data considering the three cluster strains, we decided to do the same for the 90 strains that were included in our phylogenetic analysis. Unfortunately, only 4 of the 90 strains that were included in the phylogenetic analysis actually had surveillance information. Therefore, we cannot directly relate these data to the constructed phylogeny, but we can analyze the four strains as examples of the potential of the phylogenetic–epidemiological surveillance information.

Through the Strain Name Segregation Method, we were able to draw a new picture of the host use distribution per cluster (similar to Figure 10), and we added a new column for mammalian hosts that appeared with low frequency (camels, cats, and dogs). We can see that in that sampling, the proportion of swine strains is altered compared to the main Surveillance Analysis, where the majority of strains that use this host are present in cluster 3, the majority of avian strains are from cluster 1 (which also comprises the majority of cases in general—2565 of the 4207 cases) and the majority of human strains are also from cluster 1.

Cluster 2 includes strains that mostly use swine hosts (389 of the total of 459 cluster 2 strains analyzed here), and the mammalian cases cannot be discussed here, since this group contains only 10 strains—8 belonging to cluster 3 and 2 belonging to cluster 2. As mentioned in the Materials and Methods section, however, this data does not allow us to provide any information other than strain and host group, since all we have of these extra cases is the strain name. We also could not analyze the location data provided by the strain name due to its lack of uniformity among the different strains, rendering this information obsolete (see the Appendix A).

## 4. Discussion

When applying the Stockholm Paradigm’s theoretical framework, based on the DAMA protocol described by [26], we expected to obtain, beyond the potential emerging flues, a more universal understanding of H1N1’s capacity and opportunity to emerge in humans. In order to obtain the aforementioned results, we systematically processed, merged, and analyzed data from IRD to finally statistically and visually understand the synergistic relation of this information.

Applying the MCA followed by the HCA to represent the concept of capacity described by the Paradigm [2,26], was an efficient way of representing all non-genomic information at the same time. This method allowed us to observe the 17,568 strains in a single plot, considering the information of five PCs, which cumulatively represent more than 60% of the variability present in the 6521 epitopes and the 96 genomic substitutions knowingly associated with viral capacity change, so that the strains can be organized according to their similarity (Figure 3).

The HCA grouped the strains hierarchically, generating actual clusters based on the strains’ epitope-substitution identity (Figure 3). This hierarchization was essential for us to relate the non-genomic information to our phylogenetic reconstructions and discuss the validity of how we traditionally study viral lineages or viruses’ capacity to emerge, host-switch, or alter their host-repertoire utilization (Figure 7, Figure 8 and Figure 9).

The biological understanding of the clusters, however, was provided through our subsequent analysis of the most influential epitopes given by the MCA, which shaped the work we have presented here. The simple way the MCA allows us to explore the contributions of each dimension (Table 1), as well as the parameter’s weight on the dimension, directly influenced the results presented here.

Based on the identification of the most influential epitopes, we were able to establish a systematic understanding of the biological meaning of the three clusters suggested by the HCA and bring biological meaning to the three-cluster description of the group (Figure 4). We achieved this by first establishing a general view of the inter-cluster differentiation of capacity (through the epitope frequency distribution calculation for each cluster—Figure 5 and Figure 6 and Table 2). From these plots we were able to actually see that the clusters suggested by the HCA are in fact distinct in terms of epitope profile, and also match the host use analysis (Figure 10).

The individual characterization of the ten most influential epitopes was also very important for adding depth to the understanding of the clusters. When calculating the epitope frequency of the ten epitopes (Figure 6) we were surprised by the great differentiation of the three groups, considering both the RR estimation and the pattern among the ten epitopes analyzed.

The divergence of epitope frequency among the three clusters was then understood when we traced the genomic location of the ten epitopes and discovered that they were present in the hemagglutinin (HA) genomic segment (for epitopes 1 through 5)—an envelope protein—and in the Nucleoprotein (NP) segment (for epitopes 6 through 10), a protein essential to the encapsulation of the viral segments.

It was interesting to note how the three clusters contain such divergent epitope profiles, and at such fundamental proteins of the viral structure, where the first is directly related to the viral entrance to the host cell, which has also been studied in depth when discussing tissue and disease severity (for upper or lower respiratory tract damage) and host specificity according to Sia2-3Gal (preferential for avian and equine HAs) or Sia2-6Gal receptors (preferential in human and swine HAs) [27]. Of course, this host use pattern is not absolute in terms of sialylgalactosyl tropism, but is quite homogeneous in terms of host use and disease severity, and is heavily discussed in viral hemagglutinin specificity selection (see [28,29] for the in-depth tissue tropism analyses).

The biological importance of the HA protein also leads us to hypothesize about the immunological pathway it activates, since it is an envelope protein that is constantly exposed in the host’s circulation. Based on its historical utilization in the seasonal flu vaccination programs, described as the most abundant and immunogenic protein of Influenza [30], we can understand that these epitopes interact more strongly with type B lymphocytes, and induce an antibody production according to that pathway activation [31].

On the other hand, the NP importance for host utilization also leads to important ramifications in the host–parasite immunological compatibility. Since the NP protein (and the whole vRNP) is only exposed to the immune system once the viral envelope has already fused to the endosomal membrane [27] to then establish itself in the nucleus, we can infer that these epitopes can only induce an immunological response from differentiated CD8+ T lymphocytes, except when there is an induction of adaptive immunization through vaccination [31].

The application of the MCA followed by the HCA is not innovative when discussing the simultaneous visualization and interpretation of data [32,33] or more specifically, health-related datasets [29]. It is more amply used in studies of human behavior and psychology [34] but this is the first work, to our knowledge, that discusses HCA results with an actual multi-locus phylogeny, approaching the idea that capacity might not be sufficiently explained by a single- or two-gene phylogeny.

After analyzing the single-gene trees, we constructed the multi-locus trees and compared the differentiation capacity of HA+NP and the whole-genome reconstruction, as well as how they resemble the HCA, which can be seen as a cumulative representation of “non-genomic” capacity information. This is an important step toward our goal because, when analyzing a biological group, especially in terms of their emergence risk in humans, we typically synonymize the single-gene tree with capacity to emerge, and for the HA and NP trees, capacity is different from phylogeny, or at least different from epitope-substitution information (Figure 4).

Both multi-locus trees were more similar to the HCA results compared to the single-gene trees. For both HA+NP and eight-segment trees we calculated the RR of cluster identity for the phylogenetic clusters, as an initial strategy to visualize the correlation between HCA clusters and phylogenetic groupings (Figure 7 and Figure 8). It is interesting to note, however, that although the eight-segment tree is more reliable in terms of phylogenetic resolution, it did not significantly change the clusters’ differentiation to the HCA groups, as evidenced by the small change in the relative risk calculations in Figure 7 and Figure 8 and the small change in the phylogenetic signal analysis (the phylogenetic signal analysis for the HA+NP tree is in the Appendix A).

The full-genome tree was very similar to the HA+NP analysis in terms of cluster identity, even though the eight-segment tree contains four times more genome segments for the same number of strains (Figure 7 and Figure 8). The HA+NP tree contains the two genome segments with the highest number of epitopes and substitutions that differentiate the Influenza clusters in the HCA (or non-genomic capacity information of the strains).

Although convergent, the two-segment and full-genome trees do not coincide with the capacity cluster, indicating that Influenza evolution needs to be analyzed considering other historical factors such as reassortment or recombination between strains, as well as the implications of that for the host-use repertoire. Capacity, in summary, requires more information besides gene evolution data.

For our analyzed sample, we can only understand that, for the three clusters, there is a variety of hosts being utilized, although their use did not occur in a homogeneous manner (as the capacity was not uniform among the clusters), as observed based on the host nature dataset (Figure 10b). For the three clusters, we can observe that cluster 1 contains the largest ratio of strains that utilize human hosts (more than 80% of the strains that colonize the host), whereas almost 40% of the strains that utilize swine hosts—identified as wild boars, pigs or tagged as swine in general in the dataset—are from cluster 1. This information coincides with the discussion associated with the Stockholm Paradigm, where pathogens, when exploiting a specific host (realized capacity space), still retain an ancestral capacity to explore other groups given by the sloppy fitness space of the strain, or an unused capacity in the human host which enables a host-switch when given the opportunity to do so [26].

Cluster 2, being the smallest of the three (Figure 4), but presenting the most diverse epitope frequency (Figure 5), contains strains that utilize humans, avians or swine in a homogeneous ratio. Unfortunately, only one host is attributed to each strain, since that information takes only the documented sample in which the strain was identified. Therefore, we cannot directly discuss the host repertoire of the analyzed strains [35].

When analyzing avian hosts, we can see how this host has been specified among 16 avian species (Figure 10a), which is very important when discussing the opportunity to emerge. In general terms (Figure 10b), a significant quantity of avian strains are found in cluster 3 (more than 90% of the avian cases), and almost 40% of swine Influenza strains are also from that cluster. Around 10% of strains that occur in humans are from cluster 3.

When analyzing the avian species with the highest number of records in our dataset, we can observe that both american black ducks (*Anas rubripes*) and mallards (*A. platyrhynchos*) have ample migration routes that extend from the coastal areas of North America, all through Europe and Asia [36,37]. More recently, these two species have altered their typical locations during their wintering migrations and besides being seen in coastal areas, they have explored the recently developed rural and urban localities, as their presence has been recorded in the Atlantic region of Canada [36]. Both species are traditionally monitored in various countries due to their appeal in hunting practices in all the previously mentioned regions, as well as various works related to the hypothesis of competitive exclusion dynamics between the species, as well as reproductive success changes associated with wetland quality alteration [3,38,39,40].

The American Black Duck and Mallards have a long monitoring history, and for the last 100 years they have been studied together, especially with regard to hybridization cases among the species [40,41]. The migratory pattern of ducks are usually centered in the breeding, molting, and wintering areas [42] and more recently in the mid-migratory pathways [41,43]. In Canada, both species’ migration routes have been influenced by the urbanization and agricultural degradation of forested areas, which coincides with the growing loss of wetlands due to landscape alteration in the Atlantic region of Canada [36,40,41].

It is also well documented how both species, with the loss of natural habitat, have explored urban and periurban foraging environments such as urban parks and wetlands as well as agricultural regions, which is reflected in a dietary change according to the microregion these species migrate to for the winter season [36,44,45]. What is interesting to note is that both species explore the coastal environments, except there is an ongoing discussion on how black ducks usually dominate these regions compared to mallards, due to a competitive exclusion dynamic between the two species [46,47]. The ecological importance of the black duck in all these different microregions have even culminated in the development of the Black Duck Joint Venture for a continuous monitoring of these different populations [38,48].

When discussing microregion specifications, it is very important that we differentiate the host species that were identified in our work, and how they relate to the microhabitat manually identified in our dataset. We can see from Figure 10 that all clusters were present in all six types of environments at similar ratios. Although not depicted by the plot, the location type entitled “Brackish” is the microregion with the highest number of records, and includes regions with saline water, with estuarine environments, beaches, islands, and river deltas being the specific landscapes identified in our data.

Only the “Forestal” microregion diverges in terms of cluster strains’ frequency, where in this group there is a higher frequency of cluster 2 strains, and lower presence of cluster 1 strains. Cluster 2 is the cluster with the highest diversity in terms of epitope profile and host use, whereas cluster 1 has a lower epitope frequency diversity (more homogeneous capacity profile) with a dominance of Influenza strains that occur in human and swine hosts.

We can hypothesize that this differential relation among clusters in this type of environment hints at a lower presence of humans in forested areas (although not a complete absence), and a larger distribution of avian hosts, drawn by the type of microhabitat. Forest locations have historically been described as reservoirs for various diseases, together with the human invasion of these environments being associated with the emergence of historical diseases such as HIV [34] malaria [49], zika [50], and Ebola [51], among others.

When specifically analyzing the four strains with full data, we realize that there is very little information outside of the sequence importation to Genbank, as well as the sample information that was uploaded into IRD. Our discussion will be restricted, therefore, to the data gathered in this work.

Of the four strains with both capacity and opportunity information, two are human and two are avian pathogens. Both avian pathogens utilize mallards as their main host (A/mallard/Ohio/11OS2078/2011 and A/mallard/NewBrunswick/00340/2010) and are similar to one of the identified human strains in terms of capacity (A/Managua/4905.02/2009), since the three belong to cluster 1 of the HCA. It is interesting to note how this information coincides with the distribution of the host previously mentioned with regard to its wintering distribution in the Atlantic Canada region for the New Brunswick strain (with no microregion specification), whereas the Ohio strain occurs in a preserved forest microregion, the Ottawa National Wildlife Refuge, which is also a naturally occurring microhabitat preferred by mallards.

The human utilizing Influenza that belongs to cluster 1 was identified at the Sustainable Sciences Institute in Managua, the capital of Nicaragua, which is not a traditional breeding, mid-migratory, or wintering site of mallards (or has not been surveyed for the presence of this duck species). We can hypothesize here that the human presence in these environments where mallards appear, or the recent foraging habit expansion of this host to periurban and urban areas, are enabling the continuous spillover of Influenza H1N1 between this host (and other ducks with similar behavior like their congenerics) and humans.

## 5. Conclusions

When exploring the zoonotic dynamic of Influenza A H1N1 strains, we can see how the theoretical framework presented by the Stockholm Paradigm has enabled the development of a step-by-step methodology to gather, process and interpret biological information involving pathogen–host interactions, and in this case, the capacity profile of different H1N1 groups.

The three clusters identified by the MCA-HCA procedure have allowed us to explore the multitude of datasets presented here and achieve an interpretable, straightforward result—we can rank H1N1 strains according to their capacity to utilize the human host, and we can also understand their risk of infecting humans based on their host use, geographical distribution, and ecological congruence with the human population. We also explored the phylogenetic information of the two most influential genome segments (given by the MCA-HCA), together and apart, as well as the full genome data, and compared it to the HCA results.

We realized that the three clusters presented by the HCA tend to match the phylogenetic information, meaning that the methodology presented here has the potential to complement the genomic information and enable a better understanding of a viral lineage’s capacity to infect humans by accumulating all non-genomic information to be interpreted alongside the traditional phylogenetic data.

It is important to note that Influenza is especially interesting when studying emerging infectious diseases due to its ability to reassort its segmented genome—or switch genomic segments with another coexisting strain, generating a sudden change in the viral host use ability. Reassortment was not analyzed in this work, since our primary goal was to apply the framework proposed by the Stockholm Paradigm for symbiont evolution in virus–host interactions, a method constructed that is not exclusive to Influenza. We hope this work can contribute to the public health system by supporting well-thought-out decision making regarding the emerging infectious disease crisis.

## Figures and Tables

**Figure 1 viruses-17-01350-f001:**
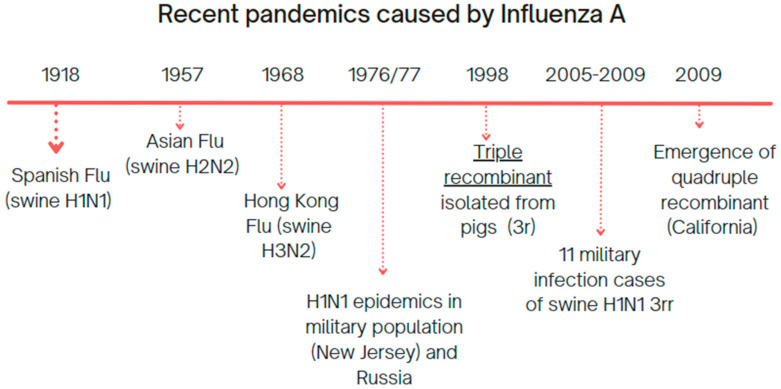
Example of Influenza A epidemic and pandemic history [12].

**Figure 2 viruses-17-01350-f002:**
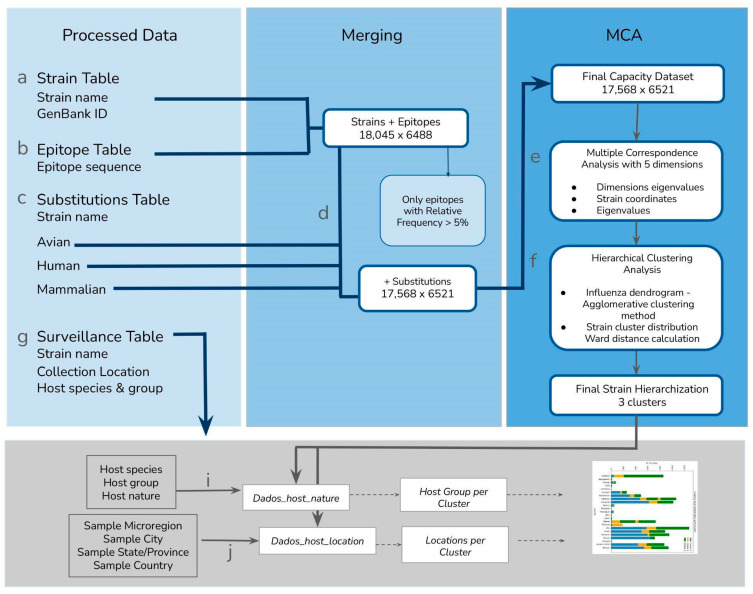
Data processing workflow—data preparation and formatting for the subsequent statistical application.

**Figure 3 viruses-17-01350-f003:**
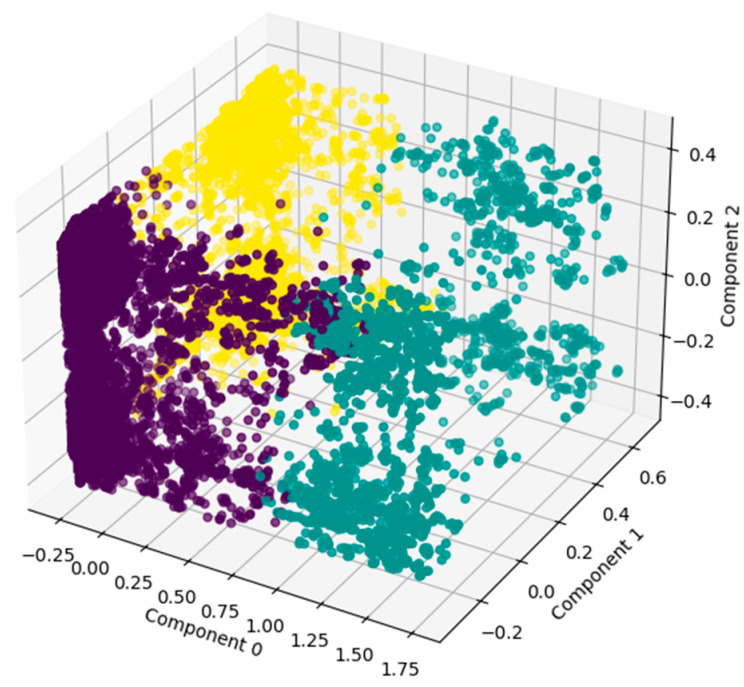
Euclidean space representation of Influenza strains colored according to the clusters from HCA. This plane representation depicts the distribution of the 17,568 Influenza strains according to all capacity parameters simultaneously, by the first three components of the MCA. Strains that are more similar according to epitopes and substitutions information will be closer to each other in the Euclidean space.

**Figure 4 viruses-17-01350-f004:**
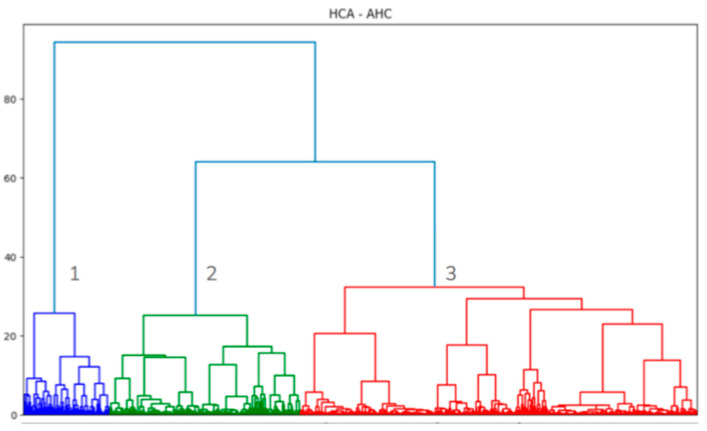
Different strain clusters according to capacity data derived from HCA. We arbitrarily and systematically (using the least vertical lines approach) divided the clusters into three groups, and will further analyze them for biological coherence.

**Figure 5 viruses-17-01350-f005:**
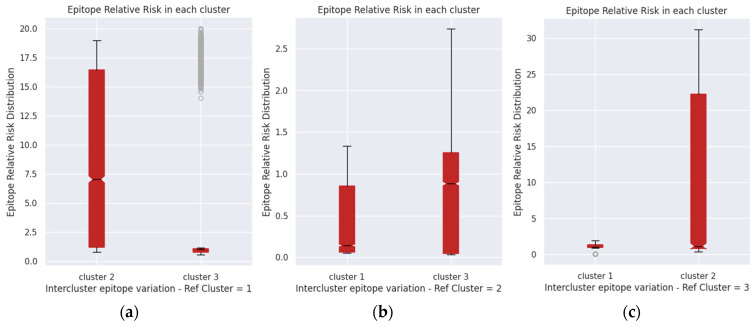
RR distribution considering, at each plot, a different reference cluster. The RR was calculated by dividing the epitope relative frequency in a given cluster by the epitope relative frequency in the reference cluster. (**a**) with cluster 1 as the reference cluster, (**b**) with cluster 2 as the reference cluster and (**c**) with cluster 3 as the reference cluster. In general terms, these graphs (this figure) can depict the epitope diversity and frequency homogeneity. Although we cannot specifically observe epitopes, the RR shows that the epitope frequency distribution is different among clusters; they present distinct epitope profiles.

**Figure 6 viruses-17-01350-f006:**
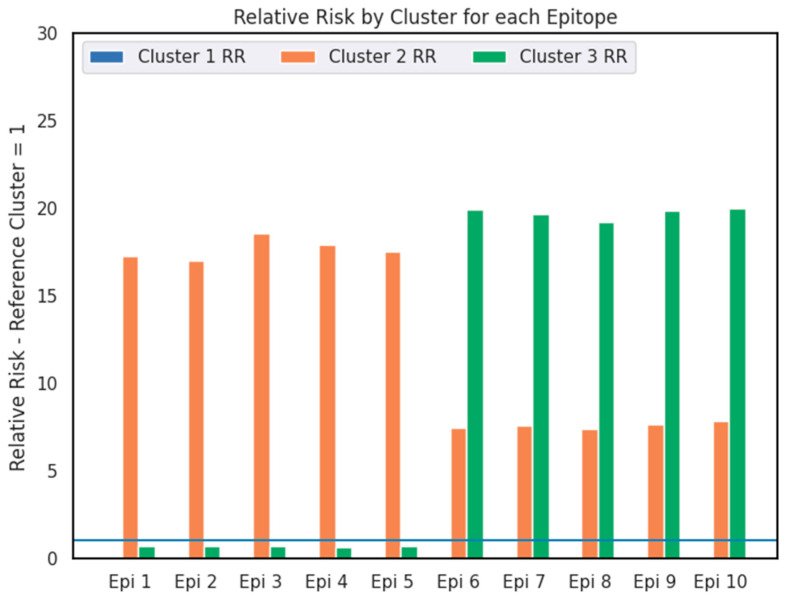
Barplot of the RR distribution for the most influential capacity parameters. The blue line represents the equality of relative frequency in comparison with the reference group (cluster 1), i.e., when RR = 1, while the orange bars represent cluster 2 RR, and green bars represent cluster 3 RR.

**Figure 7 viruses-17-01350-f007:**
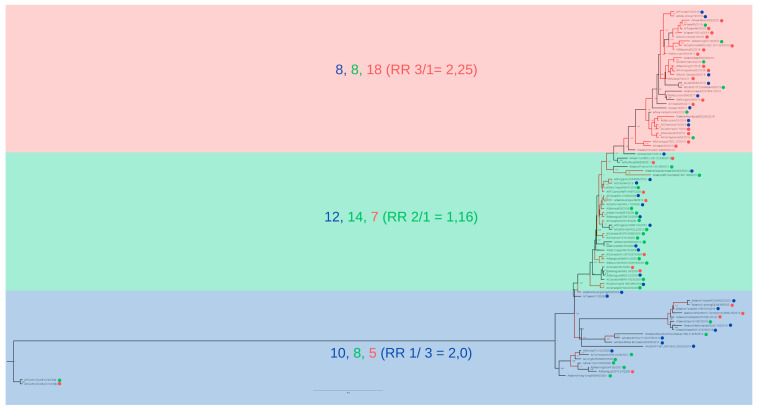
HA and NP Concatenation. Multi-locus phylogenetic reconstruction using both HA and NP segments for the 90 strains. The red branches are those with a posterior probability value under 0.5. Each strain is colored according to cluster: blue for cluster A, green for cluster B, and red for cluster C, and the tags are shaped according to the host group (circles for humans, triangles for swine, and squares for avian hosts).

**Figure 8 viruses-17-01350-f008:**
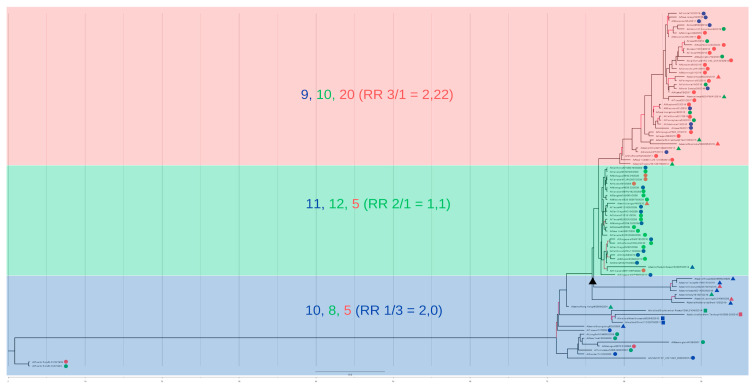
All Influenza H1N1 Segments. Multi-locus phylogenetic reconstruction using all 8 segments for the 90 strains. The red branches are those with a posterior probability value under 0.5. Each strain is colored according to cluster: blue for cluster A, green for cluster B, and red for cluster C, and the tags are shaped according to host group (circles for humans, triangles for swine, and squares for avian hosts).

**Figure 9 viruses-17-01350-f009:**
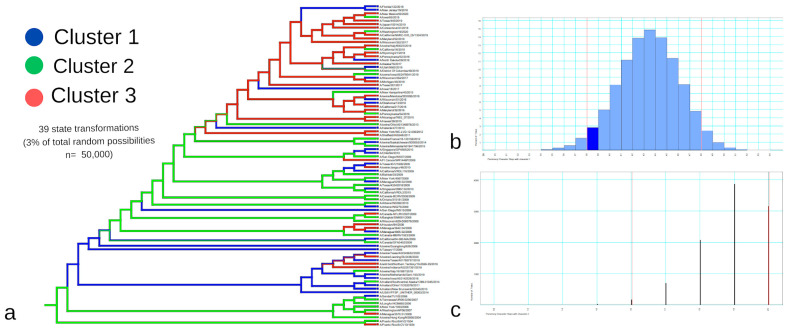
Phylogenetic signal analysis for 8-segment phylogeny. (**a**) depicts the original tree obtained in Figure 8 colored according to the cluster state evolution. (**b**) represents the phylogenetic signal calculation for the cluster identity trait, where the red lines present the 0.05 significant frequency thresholds for the number of character-state transformations, and the dark blue bar represents the frequency of 39 state transformations value (the original tree number of changes). (**c**) presents the phylogenetic signal calculation for the host use character, where the red lines present the 0.05 significant frequency thresholds for the number of character-state transformations.

**Figure 10 viruses-17-01350-f010:**
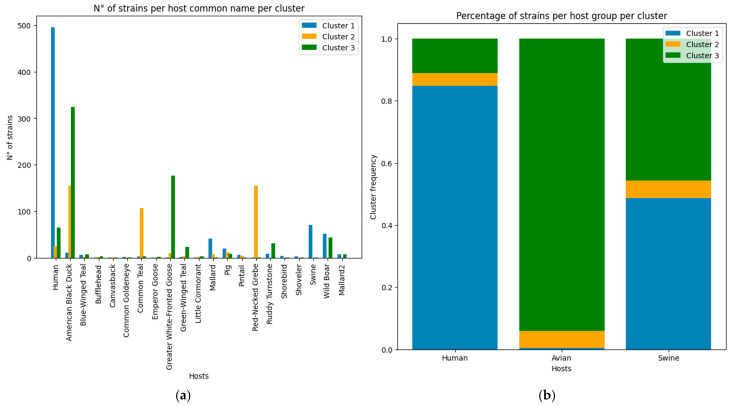
Specific host and host group count per cluster. (**a**) depicts all the information on cluster distribution according to each specific host, where the number of strains identified in the specific host is colored based on the cluster it belongs to (cluster 1 in blue, cluster 2 in orange, and cluster 3 in green). (**b**) depicts the host group distribution among clusters, where the specific hosts are grouped in human, avian, or swine types. The same colors are used to represent the respective cluster.

**Figure 11 viruses-17-01350-f011:**
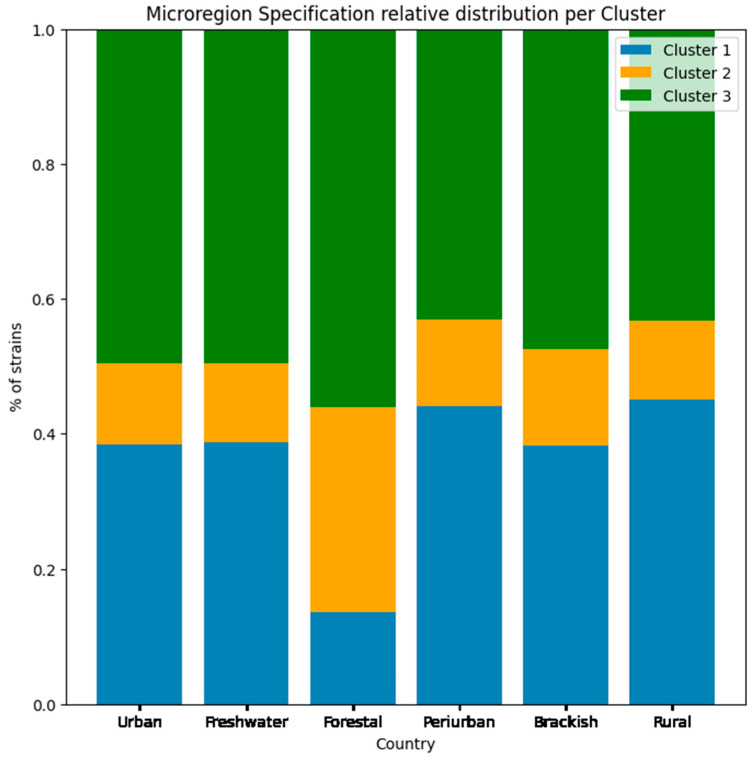
Environmental specification. Describes the relative frequency of microenvironments per cluster.

**Table 1 viruses-17-01350-t001:** Eigenvalue table for the accumulated capacity data (both epitope and substitution values).

Component	Eigenvalue	% of Variance	% of Variance (Cumulative)
0	0.292	29.30%	29.30%
1	0.142	14.31%	43.61%
2	0.077	7.77%	51.39%
3	0.076	7.61%	59.00%
4	0.056	5.61%	64.60%

**Table 2 viruses-17-01350-t002:** RR for the ten-capacity data with the highest contribution value to the first and second components, which together explain 43% of the Influenza strains’ similarity. The epitopes are described as Epi 1 through 10 in order to facilitate our interpretation (and follow this table’s order).

Epitope	Epitope Sequence	RR 2 x 1	RR 3 x 1
1	AVGREFNNLERRIENLNKKM	21.447682	0.853881
2	IATRSKVNGQSGRM	21.119145	0.818260
3	NSVIEKMNTQFTAVG	23.038797	0.797415
4	IATRSKVNGQIGRM	22.439771	0.737265
5	Y115, T148, G151, G152, S154, W170, Y172, K173, S174, G175, S176, K206, E207, N210, L211, V213, L243	21.615752	0.826451
6	KGEIRRIWRQANNG	7.805701	20.361108
7	ILRGSIAHK	7.603721	19.555801
8	SPIVPSFDM	7.382866	19.061036
9	ATEIRASCGK	7.752814	19.894004
10	VLRGSVAHK	7.927589	19.921992

## Data Availability

All raw, preprocessed, processed data and final tables, as well as the scripts and development environment used in this work, are available in this article/Appendix A. All scripts can be accessed at Laboratorio-de-Analise-de-Dados/Influenza_Mestrado (github.com).

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
