# Peer review of "Stockholm Paradigm in the Study of Influenza H1N1 Viruses: A New Approach to the Study of Zoonotic Risk Coupling Multiple Correspondence Analysis and Multi-Locus Phylogenies"

_viruses, 2025, doi:10.3390/v17101350_

Round 1
Reviewer 1 Report
Comments and Suggestions for Authors
The authors should be clear about the Stockholm Paradigm and the practical questions they want to address, which are hard to understand here.
The authors described a new way of grouping for H1N1 influenza virus and provided descriptive statistics for several characteristics. Several concerns exist: 1) the scientific questions is unclear; 2) the quantitative methodologies and logical relationship to the analyses in this study for Stochholm Paradigm is missing; 3) the biological interpretations for the methodologies and results are unclear.Author Response
We attach here our point-by-point response letter to reviewer 1.

Reviewer 2 Report
Comments and Suggestions for Authors
Recently, publications on the so-called Stockholm paradigm have begun to appear in scientific literature, and the study presented by the authors describes this Stockholm paradigm as applied to the evolution of influenza viruses of the H1N1 serotype. This is an interesting study, which, however, raises a number of questions and comments from the point of view of Virology, rather than Ecological and Evolutionary Biology.
The title of the article seems too broad and general. In the title of the article, the authors talk about a study of zoonotic risk. What viruses exactly are meant? The article only studies H1N1 viruses, which have disappeared from circulation for 16 years. All H1 influenza viruses presented in the article were originally of zoonotic origin, but once they crossed over to humans, they became human influenza viruses. Are we talking about new, potentially pandemic viruses?
The authors mentioned that they “…opted to use Influenza A H1N1 virus lineage as a study model since it has a long-lasting relationship with the human population, and has been under the scope of the global public health since the beginning of the 20th century…” (Line 62-66). However, the purpose of choosing viruses that are unlikely to ever return to circulation is unclear. If the authors aimed to test the possibility of using the Stockholm paradigm to predict future zoonotic events associated with other viruses using these H1N1 viruses as a model, this should have been stated more clearly.
If talk specifically about human H1N1 influenza viruses, they disappeared from circulation in 2009, when they were displaced by the new pandemic virus H1N1pdm09, which is still circulating. There is no evidence that non-human H1N1 influenza viruses can have any zoonotic potential. This is an evolutionary dead end. We know well the evolution of H1N1 influenza viruses from 1918 to 2009, but today these viruses are an evolutionary dead end. In the history of virology, there has never been a case of a return to circulation of pandemic viruses (the only exception is the Russian (Asian) flu pandemic of 1977, which, by the way, the WHO does not consider a pandemic). Therefore, the choice of the H1N1 serosubtype for study requires serious justification. Why did not the authors mention such a powerful tool for the emergence of pandemic viruses from zoonotic ones as reassortment? Why did the authors not mention H5 avian influenza viruses, which are currently considered the main potential zoonotic source of new emerging infectious diseases including pandemics? People started talking about the danger of bird flu viruses for humans only in the late 1990s, but there are slightly fewer publications on this topic than on the H1H1 flu viruses.
I do not consider myself entitled to praise or criticize the Stockholm paradigm, although from my point of view it is quite far-fetched and may not reflect the real state of affairs, since nature does not always fit into the framework designated by the exact sciences (mathematics, mathematical modeling, statistics, etc.) and the most obvious example of this is the absolutely uncontrolled and unpredictable evolution of influenza viruses, in particular, pandemic viruses. But I would like to get a detailed answer to the question about choosing an outdated strain as a model instead of the also comprehensively studied bird flu viruses, which are very promising in terms of zoonotic threat.
Why does an article devoted, as the title suggests, to a study of Zoonotic Risk of influenza viruses say virtually nothing about influenza pandemics, not only past ones, but also future ones?
Some authors claim that the Stockholm paradigm explains the current crisis of emerging infectious disease by integrating four elements of evolutionary biology: ecological fitting, sloppy fitness space, coevolution, and responses to environmental perturbations (doi: 10.32873/unl.dc.manter22). Others claim that it is due to (a) carbon emissions and consequent climate change; (b) resettlement/migration of people with hyper-urbanization; (c) overpopulation; and (d) human-induced distortion of the biosphere (doi: 10.3390/ijerph192416920). It seems to me that the points listed in these works differ greatly from each other. I would like to hear what exactly, in the authors' opinion, caused this crisis.
The authors pointed that "By analyzing both capacity and opportunity, we expect to be able to visualize, and hierarchize, the sampled strains according to their emergence risk in humans, contributing to emerging infectious diseases preparedness." It is completely unclear how the authors intend to transfer/extrapolate data on viruses that have disappeared from circulation to viruses that may only just appear in circulation. Based on the almost 100-year history of studying influenza viruses, one thing is known to date - their evolution has always been and will continue to be completely unpredictable.
Author Response
We attach here our point-by-point letter to reviewer 2.

Reviewer 3 Report
Comments and Suggestions for Authors
This manuscript presents a novel application of the Stockholm Paradigm to study zoonotic risk in Influenza A H1N1 strains by combining Multiple Correspondence Analysis (MCA), Hierarchical Clustering Analysis (HCA), and multi-locus phylogenetics. The study is methodologically robust and addresses a significant gap in understanding viral-host coevolution. Below are detailed comments and suggestions for improvement:
- Clarify the rationale behind excluding epitopes with less than 5% occurrence, as this may introduce bias toward conserved epitopes in the analysis.
- Discuss the marginal significance of cluster identity (p ≈ 0.05). Is this observed significance attributable to reassortment events or methodological limitations?
- Although clusters are statistically well-defined, further discussion is needed on their virological significance, specifically distinguishing between human-adapted and avian-adapted strains.
- For enhanced clarity, label workflow steps (i–j) within the main text.
- Include units for eigenvalues, such as the percentage of variance explained, to provide context.
- Avoid using jargon like "sloppy fitness space" unless it is clearly defined.
- The "Strain Name Segregation Method" appears critical but is currently relegated to the supplementary materials. Summarize the key findings in the main text for better accessibility.
Author Response
We attach here our point-by-point response letter to reviewer 3.

Round 2
Reviewer 1 Report
Comments and Suggestions for Authors
The authors have revised the text, and it is now clear. One thing the author should check is the cluster numbers used in lines 291-506; something is not right there.
Author Response
We thank you for your consideration. We attach here our response both to the correction of the cluster size description and the figure quality.

Reviewer 2 Report
Comments and Suggestions for Authors
I am satisfied with the authors' detailed explanations. I have no more comments.
Author Response
I thank you for your pertinent questions on the paper. It really enriched our work.
Reviewer 3 Report
Comments and Suggestions for Authors
The quality of the manuscript has been significantly enhanced after revision.
Author Response
We would like to thank you for your considerations on our work, it indeed improved it significantly.